# The Mechanistic Pathways of Periodontal Pathogens Entering the Brain: The Potential Role of *Treponema denticola* in Tracing Alzheimer’s Disease Pathology

**DOI:** 10.3390/ijerph19159386

**Published:** 2022-07-31

**Authors:** Flavio Pisani, Valerio Pisani, Francesca Arcangeli, Alice Harding, Simarjit Kaur Singhrao

**Affiliations:** 1School of Dentistry, Faculty of Clinical and Biomedical Sciences, University of Central Lancashire, Preston PR1 2HE, UK; 2Neurology and Neurorehabilitation Unit, I.R.C.C.S. “Santa Lucia” Foundation, Via Ardeatina, 306-00179 Rome, Italy; v.pisani@hsantalucia.it; 3Azienda Sanitaria Locale ASLRM1, Geriatric Department, Regina Margherita Hospital, Via Emilio Morosini, 30-00153 Rome, Italy; francesca.arcangeli@aslroma1.it; 4Dementia and Neurodegenerative Disease Research Group, Faculty of Clinical and Biomedical Sciences, School of Dentistry, University of Central Lancashire, Preston PR1 2HE, UK; aharding7@uclan.ac.uk (A.H.); sksinghrao@uclan.ac.uk (S.K.S.)

**Keywords:** Alzheimer’s disease, periodontal disease, *Treponema denticola*, *Porphyromonas gingivalis*, trigeminal nerve, mesencephalic trigeminal nucleus, locus coeruleus

## Abstract

Alzheimer’s Disease (AD) is a complex neurodegenerative disease and remains the most common form of dementia. The pathological features include amyloid (Aβ) accumulation, neurofibrillary tangles (NFTs), neural and synaptic loss, microglial cell activation, and an increased blood–brain barrier permeability. One longstanding hypothesis suggests that a microbial etiology is key to AD initiation. Among the various periodontal microorganisms, *Porphyromonas gingivalis* has been considered the keystone agent to potentially correlate with AD, due to its influence on systemic inflammation. *P. gingivalis* together with *Treponema denticola* and *Tannerella forsythia* belong to the red complex consortium of bacteria advocated to sustain periodontitis within a local dysbiosis and a host response alteration. Since the implication of *P. gingivalis* in the pathogenesis of AD, evidence has emerged of *T. denticola* clusters in some AD brain tissue sections. This narrative review explored the potential mode of spirochetes entry into the AD brain for tracing pathology. Spirochetes are slow-growing bacteria, which can hide within ganglia for many years. It is this feature in combination with the ability of these bacteria to evade the hosts’ immune responses that may account for a long lag phase between infection and plausible AD disease symptoms. As the locus coeruleus has direct connection between the trigeminal nuclei to periodontal free nerve endings and proprioceptors with the central nervous system, it is plausible that they could initiate AD pathology from this anatomical region.

## 1. Introduction

The literature supports that periodontal disease and Alzheimer’s disease (AD) are co-morbid [1]. This comorbidity has links with different mechanisms in which age, susceptibility genes, immunosuppression, mastication, loss of teeth, and periodontal pathogens appear to play a pivotal role. The pathological lesion of periodontal disease is the formation of ‘pockets’ typically harboring around 275 different species of bacteria [2] among which the keystone periodontal bacterium *Porphyromonas gingivalis* and oral spirochetes (*Treponema denticola*) are observed.

Since their discovery in AD autopsied brains [3,4,5,6], these bacteria have provided the trigger for exploring the potential pathways that enable their entry into the brain from their primary niche—the oral cavity. The potential influence of *P. gingivalis* and spirochetes in AD development is not in dispute [3,7,8], but the important issue is whether or not they initiate or contribute to the AD pathogenesis. Neurofibrillary tangles (NFTs), one of the two diagnostic lesions of AD, are said to spread in a predictable pattern to involve connecting neurons that project between the individual cortical regions and the hippocampus, amygdala, and association cortices of the frontal, temporal, and parietal lobes, or in reverse order from the locus coeruleus [9]. As far as the gingipain hypothesis is concerned, AD clinical trials in 2022 by the Cortexyme inhibitor molecule are supportive of potential benefits in mild to moderate Alzheimer’s cases with periodontal disease, whilst further progress is being made to correlate the benefits with a phosphorylated tau marker antibody in patients’ cerebrospinal fluid.

Thus, what are the cues that drive the migration of microbes such as *P. gingivalis* and *T. denticola* to pursue a route to the brain? Scientists have proposed putative pathways for their migration to the brain, but none have evidenced or detailed the bacterial cues that plan and prepare their journey nor demonstrate eventual pathology development and progression. This review evaluates the progress made by scientists in clarifying the pathways of bacterial entry to the brain as well as dealing with the mechanisms that support their journey, from the oral cavity to the brain.

## 2. Alzheimer’s Disease (AD)

AD is a complex neurodegenerative disease, which clinically presents with deteriorating memory, cognitive decline, along with mood changes, and it is the most common form of dementia. In AD, two histological lesions, the amyloid (Aβ) plaques and the neurofibrillary tangles (NFTs), remain the characteristic diagnostic markers at autopsy [10]. In addition to Aβ plaques and NFTs, there are other neuropathological findings (i.e., neuronal and synaptic loss, glial cell activation, and blood–brain-barrier (BBB) breach), mostly used as research parameters of pathophysiological changes [11]. The hippocampus typically contains abundant intra-neuronal NFTs composed of the abnormally phosphorylated tau protein [12]. The NFTs were thought to first appear in the entorhinal cortex, but now there is suggestion of early involvement within the subcortical nuclei such as the locus coeruleus in the pons [9,13].

Whilst Aβ plaque lesions largely serve a diagnostic purpose, the NFT lesion in AD is widely believed to correlate with disease severity, progression, and the extent of cognitive impairment [14]. Tooth loss in early and mid-life is also proposed to give rise to memory impairment in later life [15,16,17].

Our aim here is to increase our understanding of the journey that pathogens take to reach the brain and potentially influence pathophysiological events that lead to NFT formation and disease progression; an additional aim would be to determine what the lag-phase would be from initial infection to AD lesion formation. Considering that AD has a well-known, complex etiology, but with limited treatment options, new concepts and hypotheses will continue to emerge until its cause becomes established. Thus, scientists working in the microbial/aetiological aspects of AD [3,18,19,20,21,22,23] have significantly contributed to the microbial infection hypothesis of AD [24].

## 3. Periodontal Disease

Periodontal disease is an inflammatory disorder involving both the soft and the hard periodontal structures. Its pathogenesis and progression depend on the mutual interaction between bacterial infection and the host response leading to destruction of the periodontal ligament attachment and consequently triggering resorption of the alveolar bone [25]. Periodontal disease is ranked as the 6th most prevalent disease overall and it affects about 20–50% of the global population [26,27]. Its prevalence, counted as 1 in 10 adults worldwide, makes it higher and more widespread than other diseases such as cardiovascular pathologies [28,29,30]. The severe form of periodontal disease recognizes a prevalence of 11.2% in the overall population [31,32] and is on the rise, creating a significant public health concern [33].

According to the Public Health England Data, in the UK, almost 37% of the adult population suffers from moderate periodontal disease (with pockets up to 6 mm in depth), while a figure of about 9% prevalence is encountered for severe periodontal disease (with pockets deeper than 6 mm) [34].

Measuring the impact of periodontal disease on populations is one of the most challenging epidemiological observations in dentistry. The correlation of several identified risk factors at different levels (patient, tooth, and site) represents the key to a better understanding of how to develop prevention activities for the benefit of the public health and the treatment needs of the population [35,36]. This will also enable practitioners to target future interventions in terms of prevention [37]. This is especially relevant for the correlation of systemic disease with periodontal disease [38].

Several risk factors, such as smoking, hyperglycaemia, and biofilm accumulation, have been identified as playing an important role in intensifying the magnitude of the disease process via triggering an inflammatory response or negatively modulating the immune response [39]. The endpoint of periodontal disease is not simply about tooth loss, but also about an individual’s general health, correlating with other comorbid diseases including diabetes, cardiovascular diseases, and, above all, for complex, chronic neurodegenerative diseases such as Alzheimer’s disease [1,4,7,40,41,42].

## 4. The Periodontal (Sub-Gingival) Microbiome

The interaction between the oral microbiome, host response, and the oral cavity is very complex. This is mainly due to either the individual variation in composition of the subgingival flora, featuring multiple taxa of species, and/or to an individual’s immune and inflammatory responses [43]. However, bacterial dysbiosis [44,45] is the best current explanation for periodontal disease onset. The treatment of periodontal disease focuses around decreasing the pathogen load. Supporting this hypothesis is the observation that in the diseased state, there is a shift of the biofilm organisms toward Gram-negative phylotypes such as those belonging to the ’red complex’ (typified by *T. forsythia*, *P. gingivalis,* and *T. denticola)* [46]. These bacteria are associated with increased pathogenicity, severity, and resolution of inflammation in the gingival tissues [43]. According to the literature, *Treponema* species associate with the severity and pocket depth seen in periodontal disease [47]. Whether this explains the cause or the effect of the disease process remains unresolved [48]. 

Due to the selective bacterial differences of the sub-gingival periodontal disease microbiome compared to other oral diseases [49], for example, caries, it could be theorized that the development of this diverse consortium of bacterial communities seen in periodontal disease may be related to a larger scale of metabolic interactions among the multispecies biofilm consortium. This would be in conjunction with supporting an impaired immune response and/or by increasing levels of glycoproteins in the gingival crevicular fluid [50] along with a 4-log increase in the total microbial biomass [51]. In addition, *P. gingivalis*, with its keystone bacterial properties, serves to disable and deregulate effects on the local bacterial residents and the host immune and inflammatory systems via its virulence factors (lipopolysaccharide—LPS, proteases—gingipains, and adherence/motile appendages—fimbriae), which, together, potentially act as the main orchestrators of the observed dysbiosis [45,52]. As a result, this has the potential to incite organ-specific inflammation via increased levels of IL-1β, IL-6, TNF-a, chemokines and IL-8 and, in the case of the brain, to exert negative effects on the permeability of the BBB [53]. The burden of *P. gingivalis* and *T. denticola* is strongly related to observed clinical periodontal indices, namely an increased Probing Pocket Depth, Clinical Attachment Loss, and overall severity of periodontal disease [46,54,55,56].

A complex mutual interaction between these taxa has been highlighted in many experimental studies [57,58], whereby *T. denticola* can reduce the numbers of *P. gingivalis* cells initially required for periodontal disease development [59] and facilitate not only the adherence of the keystone bacterium to the biofilm early colonizers such as *Streptococcus gordonii* [58], but also to the host’s epithelial cells via enhancing the expression of Arg-gingipain A (RgpA), Lys-gingipain (Kgp) gingipains, and hemagglutinin A (HagA) [60]. Conversely, *P. gingivalis* can aid *T. denticola* to regain its spiral morphology and support its motility throughout the polymicrobial biofilm [61], within the highly viscous environments, the tissues, while simultaneously creating pores to improve permeability and diffusion of the chemical stimuli and to maintain cell-to-cell communication [62]. The periplasmic flagellum of *T. denticola* is composed of three parts (a basal body, hook, and filament) to aid its motility [63]. The other extracellular flagellum aids the spread of the microbe in every cellular milieu including the cell cytoplasm [64].

## 5. The Enteric Nervous System

Historically, it is the central nervous system (CNS) that is reportedly affected by the pathophysiology of AD. However, the human microbiome project (2010) strongly suggests that gastrointestinal tract (GI) dysbiosis is also associated with pathogenic mechanisms in AD [65]. The enteric nervous system (ENS) constitutes sensory, motor, and interconnecting neurons from the esophagus to the rectum as part of the autonomic nervous system. It has links with the CNS via the brain stem and the vagus (10th cranial) nerve. The vagus nerve terminates in the nucleus of the solitary tract (also known as the organ of stress) in the brain stem. The literature has reported the presence of Aβ plaques in the intestinal submucosa of two AD patients [66]. This observation led to the suggestion that AD brain pathophysiology may begin in the ENS, with Aβ acting as a transferable seed. This is a plausible hypothesis as microbiologists have suggested that insoluble human Aβ equivalents are found in bacterial biofilm matrices where they serve as functional amyloids (acting as pillars to support a three-dimensional biofilm structure) and, consequently, offer protection [67]. Indeed, phylogenetic analyses of 16S rRNA gene sequencing studies show that the AD brain(s) overall contains more bacteria than their age-matched control autopsy brain(s) [6,68,69] reported up to a seven-fold increased density of oral bacteria in AD brains compared to the age-matched control (non-AD) brains. Therefore, the gut–brain axis may also contribute to the bacteria observed in AD brains [70] as Gram-positive cocci and rod-shaped bacteria reside in the mouth, in the gastrointestinal tract, and in the brain. Of these, Gram-negative rod-shaped bacterial genera *Porphyromonas* [3], *Actinomyces*, *Prevotella,* and *Treponema* have been identified within autopsied AD brains [6,68,69]. In addition, the spirochete *Borrelia burgdorferi* [21] was identified in the Emery’s [68] investigation. Spirochetes such as *Treponema* species have been linked to AD for at least two decades [6] and increased levels of lipopolysaccharide and DNA to *P. gingivalis* [3,5], and increased levels of serum immunoglobulins to *Fusobacterium nucleatum* and *Prevotella intermedia* have all been associated with cognitive impairment and/or with AD [71]. Recently, there has been consideration toward the marked neurotropism of spirochetes and the central role of the trigeminal pathway among different peripheral nerves. Neuroanatomical evidence has demonstrated an intimate proximity between the trigeminal nuclei and the locus coeruleus, as well as a direct connection between the periodontal free nerve endings and proprioceptors within the central nervous system.

## 6. The Trigeminal Connections between Periodontal Ligament and the Limbic System

Recently, the USA pharmaceutical company Cortexyme has focused on the role of *P. gingivalis* in AD, investigating the potential impact, via inhibition of gingipains, with encouraging results. *P. gingivalis* has been shown to enter the bloodstream via bacteraemia-based leakage from the periodontal pockets, and this implicates its role in neuroinflammation and amyloid deposition [53,72]. From these oral connections, reports have highlighted that *P. gingivalis* may reach the CNS more rapidly due to the contiguity of the anatomical structures from the upper molars via the maxillary sinuses. However, experimental and/or primary evidence is lacking in detecting *P. gingivalis* from the olfactory nerve [73]. Theoretically, the idea is plausible as most of the primary afferent neurons related to cranial somatosensory function and oral stereognosis are related to the trigeminal (V) cranial nerve. The central processes of the Gasser’s cells enter the pons via the sensory nerve root and form a descending bundle or tract that lies in the dorso-lateral region of the brain stem. It extends from mid-pons up to the 2nd and 3rd cervical cord segment overlapping with the Lissauer’s tract [74]. The somatotopic organization of the bundle appears to correlate more with the medial mandibular nerve fibers, the lateral nerve fibers, and with the ophthalmic nerve. The brain stem nuclei include the principal nucleus and the spinal trigeminal nucleus, which, in turn, comprises the subnuclei caudalis, the interpolaris, and the oralis according to a caudal-rostral location [75]. The mesencephalic trigeminal nucleus (Vmes), which includes a mixture of pseudo-uni and multi-polar cells, extends from the pons to midbrain. The multipolar cells project to rostral areas such as the locus coeruleus, and the pseudo-unipolar cells contain afferents from the peripheral areas such as the periodontal tissues [76,77]. Neuroanatomical studies have clarified that the periodontal ligament surrounding teeth contain a variety of neuronal sensory endings [78,79]. Low-threshold mechanoreceptor coding for tooth displacement has shown their cell bodies to be in the caudal portion of the Vmes [80] as well as in the posterolateral portion of the Gasser’s ganglion (both the mandibular and the maxillary division) [81]. Few rapidly adapting afferent fibers from the periodontal proprioceptors are found to project directly to the V Motor Nucleus without any stop-over in the Gasser ganglion [82]. Recently, it has been suggested that early AD may develop first in the locus coeruleus [83], and its overall volume decreases according to the Braak Staging [84]. The locus coeruleus, which innervates several regions of the brain, is the primary source of norepinephrine in the CNS. Norepinephrine is a major modulator of behavior, offering neuroprotection and suppressing neuroinflammation. Damage to the locus coeruleus may dysregulate norepinephrine release, which would be detrimental to brain function [85]. Anatomically, the locus coeruleus is closely located adjacent to the Vmes in the lateral part of the periaqueductal grey matter of the fourth ventricle. Therefore, the locus coeruleus and Vmes may negatively impact on each other, with neurodegeneration as the consequence [86,87]. This theory is supported by clinical evidence that associates tooth loss and cognitive decline [88], and experimental research has demonstrated that molar extraction in transgenic mice may result in neuronal death in the Vmes due to the axonal damage of periodontal afferents, and the spread of cytotoxic Aβ_42_ to the locus coeruleus and, from there, to the hippocampus, with consequent cognitive decline [89]. Nevertheless, it is interesting to mention that a true neural connection between Vmes and the locus coeruleus has been identified: a few synaptic bouton-like swellings appeared to contact the Vmes ganglion-cell bodies, whereas others were distributed in the Vmes region without apparent synaptic contacts [86]. However, the proximity of both anatomical structures may enable the cytotoxic effect of Aβ_42_ to reach the locus coeruleus together with the inflammatory effect on the microglia [89] (see Figure 1 for the anatomical pathways).

## 7. Could the Trigeminal Nerve Pathway Act as the Entrance for *T. denticola* into the Brain?

Experiments on mice have shown *T. denticola* as not having a secondary role in neurodegeneration. This oral spirochete can enter the brain and it has been detected in the trigeminal ganglia and in the hippocampus [6]. An experimentally induced infection of the oral cavity with *T. denticola* demonstrated the release of Aβ_1–40_ and Aβ_1–42_ from its parent protein (amyloid precursor protein) due to the activation of β-secretase and γ secretase [90] and resulted in the promotion of GSK-3β activation and tau phosphorylation [8]. 

Miklossy in 2015 reviewed the evidence from human specimens in AD and syphilitic dementia. General paresis and other chronic spirochetal infections share with AD the same local amyloidosis and the same cognitive decline; however, their presence in the body could occur years or decades before the dementia has become symptomatic or evident. The Gasser ganglion, according to Riviere, shows a massive accumulation of spirochaetes [6,91]. The potential for these oral bacteria to contribute to AD lesion formation is not in doubt, but is there a lag phase before their contribution to AD lesions becomes more obvious? One plausible reason for a lag phase could be that spirochetes are very slow growing in the brain after their translocation from their primary niche. The complex dysbiosis in periodontitis and the synergistic interaction between *P. gingivalis* and *T. denticola* may encourage the centripetal pathway of the motile *T. denticola* along the trigeminal nerves from the deep compound of the periodontal ligament, where the main proprioceptors and the free nerve termination are located [92].

As with other spirochetes (*Borrelia burgdorsferi*), *T. denticola* may migrate along the peripheral nerves or lymphatic vessels to the central nervous system as evidence shows that the spirochetal chemokine CXCL13 can be found in high concentrations in the cerebrospinal fluid but not in the serum [93].

Along the described afferent pathways, *T. denticola* may directly reach the Vmes with or without interfacing at the Gasser ganglion, triggering an initial neuroinflammation process that extends to the locus coeruleus. The local amyloid deposition in the locus coeruleus and the norepinephrine derangement could explain the initial mild cognitive impairment and the early AD symptoms due to hippocampal diencephalic and para-hippocampal retro splenial network involvement [94]. A similar process could involve the entorhinal cortex as many studies on trigeminal neuralgia are showing an overall gray matter volume reduction in patients with chronic diseases [95]. The increasing burden of *P. gingivalis* and the inflammation and leakage of inflammatory mediators from the established periodontal disease lesions may initiate the latter effects of gingipains in the alteration of BBB permeability [53]. The migration of *P. gingivalis* to the brain via the blood circulation may involve the temporal-amygdala-orbitofrontal network, which could explain the advanced signs and symptoms of late AD and dementia [94]. In this scenario, *T. denticola* may be transporting *P. gingivalis* to the brain when a plateau of either amyloid deposition or microglial inflammation has already been established for its survival and sustenance to facilitate its effect on different neuronal circuits.

## 8. Discussion and Implications for Future Research

Along with the many different etiologic factors contributing to AD, such as genetics, diabetes, hypertension, metabolic syndrome, and cerebrovascular disease, bacterial dysbiosis appears to be related [96]. An alteration of the symbiotic relationship between the biofilm-related microbial community and the host may lead some selected species to trigger a local persistent inflammatory response and an endothelial dysfunction with a consequent dissemination of pathogens to distant organs, such as the brain. This is in keeping with the theory of an infectious etiology of AD, based on the common specific hallmarks of neuronal loss, progressive synaptic dysfunction, deposition of amyloid-β (Aβ) peptide, and the abnormal forms of tau protein [20]. The role of oral bacteria in this perspective is quite evident, triggering an acute liver inflammatory response via IL6, characterized by increased pro-inflammatory cytokines. These exert their impact from a distance or, locally, following entry into the brain, or may even be released locally, influencing the brain function [97]. Some cytokines can reach the brain, despite the BBB functioning to prevent noxious substances accessing the CNS. Macromolecules, such as LPS, can enter the brain by passing from the circumventricular organs’ capillaries where the BBB is lacking, or by using specific transporters that increase BBB permeability [98]. It has been clarified that BBB breakdown is always associated with cognitive decline and AD [99]. Despite the strong evidence from laboratory-based research [8,90] and the histopathological observations [6], the role of periodontal spirochaetes, and indeed *P. gingivalis*, in the pathogenesis of AD is widely underappreciated among the scientific community. The synergism between *T. denticola* and *P. gingivalis* in the oral subgingival dysbiotic biofilm may enable the former to regain its motility and move along the neural cells in the surrounding disrupted periodontal tissue via proprioceptors and the trigeminal free nerve endings. Bypassing any BBB obstacle (not present at the trigeminal level), *T. denticola* could directly reach the Vmes and locus coeruleus, initiating an inflammatory process and, consequently, neurodegeneration as well as norepinephrine imbalance. As soon as the established periodontal lesion induces and activates a systemic inflammatory response via the cytokine cascade effect, *P. gingivalis* would be able to enter the bloodstream via the inflamed periodontal tissues [38]. From there, entry into the brain would be possible via the ensuing increased BBB permeability, contributing to the neurodegenerative process from the pons and the locus coeruleus. This sequence of pathogenetic effects would enable justification and explanation of the temporal lag phase from the onset of mild cognitive impairment to dementia onset and would help to explain the differences observed in the initial clinical signs and symptoms compared to the late phase of dementia. Whether or not the actual neurodegeneration of Vmes and locus coeruleus cells could be uniquely attributed to tooth loss (a true endpoint in periodontal disease), or to a potential combination of an early *T. denticola* infection and the later deafferentation effect, represents an interesting aspect warranting further investigation. Further research would be required to investigate the major role of the trigeminal connections with the limbic system, and how the potential effect of periodontal treatment and supportive therapy to stabilize periodontal disease progression may slow the progression of cognitive decline.

## 9. Conclusions

The neuroanatomical evidence shows an intimate proximity between the trigeminal nuclei and the locus coeruleus, as well as a direct connection between the periodontal free nerve endings and proprioceptors with the CNS. Spirochetes are part of the red complex consortium of bacteria with mutual synergy. *T. denticola* (as an example of a spirochete) is the larger bacterium upon which *P. gingivals* could easily hitch a ride to enter the CNS due to the marked neurotropism displayed by spirochetes. In addition, the trigeminal pathway is not subject to the BBB, thus proving easy access to the brain via an alternative route. The proposed pathway is likely to circumvent the initial immune recognition of these bacteria as they will hide within the ganglia, perhaps for many years. From here, they may invade the neighboring neuroanatomical areas of the brain, areas such as the locus coeruleus, affecting neurotransmitter release, leading to early depressive signs in the host. The eventual spread of these bacteria into other areas of the brain affected by AD will cause neuroinflammation, encourage hallmark lesion formation, and affect the permeability of the BBB to allow more oral bacteria including *P. gingivalis* to enter the CNS via the bloodstream. Thus, slow-growing bacteria with the ability to hide within ganglia for many years, and evasion of the hosts’ immune responses may explain the extended lag phase observed between lesion formation and the eventual diagnosis of this neurodegenerative disease. If there is relevance of this potential association with oral bacteria to AD cause, the importance of maintaining good oral hygiene to keep the numbers of these bacteria low and within their primary niche is of utmost importance.

## Figures and Tables

**Figure 1 ijerph-19-09386-f001:**
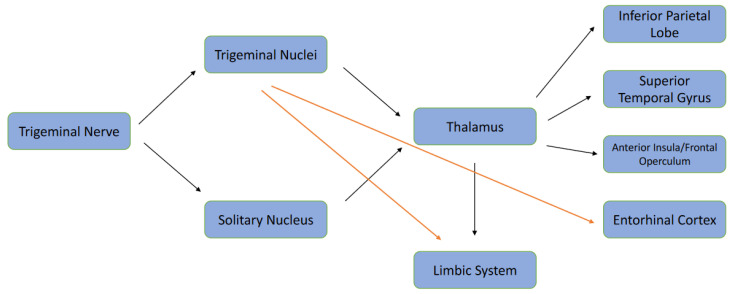
Trigeminal Neuro-anatomical Connections. Black Arrows: Established Pathways. Red Arrows: Hypothetical Pathways.

## Data Availability

Not applicable.

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
