# Peer review of "The Mechanistic Pathways of Periodontal Pathogens Entering the Brain: The Potential Role of Treponema denticola in Tracing Alzheimer’s Disease Pathology"

_ijerph, 2022, doi:10.3390/ijerph19159386_

Round 1
Reviewer 1 Report
The Authors presented a significant problem of the relationship between periodontitis and general diseases. This work is of interest to periodontology as well as neurology.
However, I would encourage the Authors to pay attention to the following:
Line 15- a the most common ?
24- . after T
Abstract: The Authors should add conclusions.
Line 39,43: Spirochetes or spirochetes( is capital letter correct?)
115- “Several risk factors have been identified as playing an important role in intensifying 115
the magnitude of the disease”- what are the risk factors? Authors should explain better the topic.
125- rather oral cavity than mouth
Reviewer 2 Report
The review explores the interesting topic of mechanistic insights about oral bacterial involvment in AD processes. Although a well-defined purpose and a clear argument of investigation, some major issues need to be raised.
1) authors should define what kind of review they were conducting.
2) authors provide a detailed dissertation of neuroanatomical links between oral cavity and AD neuronal key neuronal spots, inherently suggesting this link as a main pathway between oral dysbiosis and clinical evidence of bacteria in AD brain; however, how bacteria such as P. gingivalis and T. denticola can spread along nerves, and the factor stimulating or inhibiting this pathway is not discussed
3) authors seems to support a distinct way of brain colonization: T. denticola via nerve and P. gingivalis by blood; are they sure of these strictly distinct pathways?
4) P. gingivalis is greatly discussed into this review, enough to deserve about the same importance of T. denticola and thus the presence into the manuscript's title.
5) the only figure is a scheme of neuroanatomical connections; it would be useful and in line with the topic of the manuscript to have at least one figure on pathogenic mechanisms of oral bacteria to AD.
6) the potential role of periodontopathogenic extracellular vesicles in promoting neuroinflammation and neurodegenration should be somewhere discussed
7) Some recommended readings to improve the work include:
doi: 10.1007/s00784-020-03764-w
doi: 10.3390/biom11060845
doi: 10.4103/1673-5374.310672
doi: 10.1038/s41577-020-00488-6
doi: 10.1155/2020/2146160
doi: 10.3389/fmolb.2020.596366
Specific comments
8) Lines 15: reword "a the most"
9) Lines 51-54: add references (registered trials, website...)
10) Line 137: please reword
11) Lines 202-206: add references
12) Lines 296-299: maybe trojan horse is not the right definition here, unless authors provide evidence that brain interpret T. denticola as a "good or interesting" factor
13) Overall: please be consistent in the use of italic when reporting bacteria
Round 2
Reviewer 2 Report
Accept in present form.